# Expression of Tumor Suppressor SFRP1 Predicts Biological Behaviors and Prognosis: A Potential Target for Oral Squamous Cell Carcinoma

**DOI:** 10.3390/biom12081034

**Published:** 2022-07-27

**Authors:** Chun Chen, Yifei Zhang, Yupeng Liu, Lei Hang, Jun Yang

**Affiliations:** 1Department of Otorhinolaryngology-Head & Neck Surgery, Xinhua Hospital, Shanghai Jiao Tong University School of Medicine, Shanghai 200092, China; acbbaz@sjtu.edu.cn (C.C.); zhang.yf@sjtu.edu.cn (Y.Z.); liuyupeng@xinhuamed.com.cn (Y.L.); 2Ear Institute, Shanghai Jiao Tong University School of Medicine, Shanghai 200092, China; 3Shanghai Key Laboratory of Translational Medicine on Ear and Nose Diseases, Shanghai 200092, China; 4Business School, Tianhua College, Shanghai Normal University, Shanghai 201815, China

**Keywords:** oral squamous cell carcinoma, SFRP1, tumor infiltration, invasion, prognosis

## Abstract

Background: Genomic instability is implicated in the initiation and progression of oral squamous cell carcinoma (OSCC). Tumor suppressor Secreted Frizzled-Related Protein 1 (SFRP1) may participate in the aberrant evolution of OSCC, the intrinsic molecular mechanisms of which may provide effective therapeutic targets. Methods: A bioinformatics analysis was carried out on a publicly available database using R language to map the prognostic value, immune infiltration and enrichment of SFRP1 expression. Subsequently, in vitro experiments were conducted to unveil the biological function of SFRP1. Results: SFRP1 was found to be ubiquitously lowly expressed in OSCC using a Wilcoxon rank-sum test. Univariate analysis confirmed that those patients characterized by a low SFRP1 expression were significantly associated with advanced T-stage, clinical stage and poor mortality (*p* < 0.05). Furthermore, SFRP1 displayed a positive performance in tumor immune infiltration, especially in mast cells. Functional annotations indicated that highly expressed SFRP1 was associated with membrane potential and passive transmembrane transporter activity and it was mainly enriched in calcium pathway and neuroactive ligand–receptor interaction. In vitro, the overexpression of SFRP1 inhibited its proliferation, migration, and invasion and resulted in G0+G1 phase arrest within Cal27 cells (*p* < 0.05). Conclusions: The bioinformation data suggest that SFRP1 expression provides an insight into the risk and prognostic stratification in OSCC. SFRP1 was validated as a potential biomarker with anticarcinogenic behaviors for use in targeted therapy.

## 1. Introduction

Oral squamous cell carcinomas (OSCC), the most prevalent subgroup of head and neck squamous cell carcinoma (HNSCC), accounts for 3.5 million diagnosed neoplasia annually [1,2]. Dining habits such as tobacco and alcohol consumption contribute to high incidence rates globally; however, the 5-year survival rate of advanced OSCC was only estimated at 34% with multimodal treatments [3]. Despite the increasing awareness of the clinicopathologic features and molecular mechanisms of OSCC, multimodal treatment outcomes have not been optimized significantly; these largely depend on chemo-resistance, long-term radiation toxicities, locoregional relapse and neck recurrence [4,5]. Due to the abundant lymphatic drainage of special locations, upon infiltration into the lymphatic vessels, tumor cells extravasate from the vessel walls and start a proliferative program, then they detach from the primary to the surrounding extracellular matrix and distant metastases [6,7]. It has been reinforced that the prediction of tumor progression and recurrence mainly relates to clinical stage, histologic differentiation, depth and pattern of invasion, nodal burdens and metastases. Importantly, the progress from normal mucosal dysplasia to squamous cell carcinoma is a multistep process that involves polygenes. Hence, the identification of potential targets and regulatory pathways-induced OSCC is imperative to facilitate early diagnosis and improved prognosis.

The discovery of a tumor suppressor marks a new field in the research of guiding tumorigenesis. Secreted Frizzled-Related Protein 1 (SFRP1) is an important protein-coding gene belonging to the SFRP family. The mutation or epigenetic alteration of SFRP genes renders them highly susceptible to Wnt-regulated activities. The down-regulation of SFRP1 has been observed in a variety of solid tumors such as gastric cancer [8], colorectal cancer [9], etc., suggesting that the inactivation of SFRP1 is an essential prerequisite for cell differentiation and carcinogenesis [10]. However, our current understanding of the underlying mechanisms of SFRP1 is still very limited and its molecular function in oral cancer cells is yet to be revealed. So far, no reproducible indicators of an SFRP1 signature in OSCC have been reported.

In this study, we aimed to elucidate the relationship between SFRP1 expression and the oncological outcomes of OSCC. To the best of our knowledge, this is the first analysis that combines SFRP1 bioinformation and in vitro experiments from previously published research. This study investigated the influence of various clinical characteristics, including prognostic analysis, immune infiltration, and enrichment of biological functions and pathways, using the data from The Cancer Genome Atlas (TCGA). Lentivirus transfection, Western blotting, transwell assay, wound-healing assay and flow cytometry methods were used to verify the relationship between SFRP1 expression and the biological behaviors of OSCC cells to provide further information about the adjuvant regimens and precise approaches used in treating OSCC.

## 2. Materials and Methods

### 2.1. Acquisition of Single Gene Matrix

The SFRP1 gene expression profile matrix and the corresponding clinical information of 32 normal and 329 OSCC tumor tissues (HTSeq-FPKM data) were downloaded from the official website of TCGA (https://tcga-data.nci.nih.gov/tcga/, accessed on 10 June 2022). A flow chart of the data analysis and the experimental design is shown in Figure 1. R software (version 3.6.3; https://www.r-project.org/, accessed on 10 June 2022) was used to standardize the RNA sequencing and FPKM data. Differences in the SFRP1 expression levels of tumor tissues and pan-cancer tissues were analyzed with the edgeR software using a Kruskal–Wallis test. The “ggplot” package was used to create box plots.

### 2.2. Tumor Immune Infiltration Analysis

The “GSVA” package was used to investigate the relationship between SFRP1 expression and immune cell infiltration. Based on the signature genes obtained for various immunocytes, the comparative enrichment score of each type of immune cell was quantified from the gene expression profile [11]. The correlation of 24 types of immune cells was calculated using Spearman’s analysis, which was declared significant when *p* < 0.05. In addition, we constructed scatter diagrams to observe the relationship between SFRP1 expression and the six most significant immune cells.

### 2.3. Prognostic Analysis

To identify the clinical significance of SFRP1, the primary end point of this study was overall survival (OS). The “survivalROC” package was used to produce AUC values. The relationship between SFRP1 expression and OS was investigated using a univariate analysis. To compare OS with different levels of SFRP1 expression, Kaplan–Meier analyses and curves were employed, with a log-rank of *p* < 0.05 being considered significant. The R “rms” package was used to create a nomogram chart; subsequently, calibration and visualization were used to assess the constructed nomogram.

### 2.4. Enrichment Analysis

The Search Tool for the Retrieval of Interacting Genes (STRING; http://string-db.org, accessed on 22 July 2022; version 11.5) was applied to describe the protein–protein interaction (PPI) network of SFRP1. The high-SFRP1-expression group and low-expression-group were compared to screen for differentially expressed genes (DEGs) using the “limma” package. Gene Ontology (GO) and Kyoto Encyclopedia of Genes and Genomes (KEGG) were applied to uncover the significant functional and pathway differences between the high- and low-expression groups in the DAVID database (https://david.ncifcrf.gov/, accessed on 22 July 2022). GO analysis was performed using the “ClusterProfile” package based on three categories: biological processes (BP), molecular function (MF) and cellular component (CC). Differences with |log fold change (FC)| > 1.5 and an adjusted *p*-value < 0.05 were considered threshold values.

### 2.5. Cell Culture and Lentiviral Transfection

To leverage the bioinformation analysis, oral carcinoma cell line Cal27 (China Center for Type Culture Collection, Shanghai, China) was maintained in a culture medium containing 10% fetal bovine serum (FBS, Natocor, Córdoba, Argentina) at 37 °C in a humid atmosphere with 5% CO_2_. The recombinant lentivirus named GV493 was utilized to knock down the overexpressed SFRP1. The lentivirus was provided by Jikai Gene Biological Inc. (Shanghai, China). All transfection experiments were performed in accordance with the protocol of the manufacturer. The result of transfection was observed under a fluorescence microscope. The SFRP1 expression was quantified through a reverse transcription-polymerase chain reaction (RT-PCR) and a Western blotting assay.

### 2.6. Western Blotting Assay

The cytoplasmic proteins of Cal27 cells were lysed in RIPA buffer (Beyotime, Beijing, China). The lysates were treated with 10% SDS-PAGE gel and transferred onto PVDF membranes (Invitrogen, NY, USA). After 1 h of incubation, the PVDF membranes were cultured with anti-SFRP1 (ab126613, Abcam, 1:3000) antibodies overnight at 4 °C. After PBS washing, the membranes were incubated with a secondary antibody (Beyotime) at 37 °C for 1 h. The reaction was visualized and quantified using the ImageJ software.

### 2.7. RT-PCR

To further verify the result of the lentiviral transfection, total RNA was extracted from the Cal27 cell using Trizol Reagent and reverse transcribed using the RT reagent kit gDNA Eraser (Takara, Tokyo, Japan). Primers were shown as follows: SFRP1 forward: 5′- AGTCGGACATCGGCCCGTAC -3′, Reverse: 5′-AGTCGGACATCGGCCCGTAC-3′; GADPH forward: 5′-TGACTTCAACAGCGACACCCA-3′, Reverse: 5′-CACCCTGTTGCTGTAGCCAAA-3ʹ. Every sample was examined three times. The quantitative calculation was performed using system software and the 2^−ΔΔCt^ method.

### 2.8. Cholecystokinin-8 (CCK8) Assay

The CCK-8 assay was utilized to measure cell viability 24 h, 48 h and 72 h after transfection. Cells were seeded onto 96-well plates with a mixture of 100 l medium and 10 l Cell Counting Kit8 (CCK8, Biosharp, Hefei, China). The optical density (OD) at 450 nm was recorded to measure the cells’ proliferation ability after 1 h of incubation.

### 2.9. Transwell Assay

The invasive ability of the Cal27 cell was assessed using the chambers of 24-well plates. In the upper chamber, Cal27 cells were planted in the Matrigel matrix (356234, BD Sciences) and a serum-free medium. In the lower chamber, full culture medium was supplied and the chambers were incubated for 24 h. The average number of migrated cells in each group was counted under a microscope at 200× magnification. All of the experiments were conducted in duplicate and repeated three times.

### 2.10. Wound-Healing Assay

A wound was scratched into the cell monolayer; subsequently, the photos were taken every 4 h from 0 to 24 h after the wounding at the same location. After 24 h, the migrated Cal27 cells to the scratched area were calculated using Image-Pro Plus 6.0.

### 2.11. Flow Cytometry Assay

Cell cycle distribution was measured using a cell cycle detection kit (KGA512, KeyGEN, China). The cells were fixed in 70% ethanol overnight at 4 °C, washed, and suspended in a 0.5 mL solution containing propidium iodide and RNaseA (Sigma Aldrich, St. Louis, MO, USA). They were examined using the ModFit software after 45 min of incubation at room temperature.

### 2.12. Statistical Analysis

Measured data were described using mean ± standard deviation (SD). The R software (version 3.6.2) was used to realize the Kaplan–Meier curves and log-rank tests. The chi-square test was performed using IBM-SPSS version 25.0 to evaluate the relevant clinical features. A one-way analysis of variance (ANOVA) was used to compare data from several groups. We repeated all the experiments at least three times. *p* < 0.05 was considered statistically significant (* *p* < 0.05, ** *p* < 0.01, and *** *p* < 0.001).

## 3. Results

### 3.1. SFRP1 Is Down-Regulated in OSCC Tissues

To summarize the level of SFRP1 expression in pan-cancer and normal tissues, the Kruskal–Wallis test was performed, finding that the SFRP1 expression levels were significantly lower in most solid tumors than in paired normal tissues (*p* < 0.05, Figure 2a). Relevant studies of OSCC that evaluated the expression level of SFRP1 showed that it was dramatically down-regulated in tumor samples compared to control ones (*p* < 0.001, n = 32 versus n = 329, 3.67 ± 2.10 versus 5.91 ± 1.57), which suggested a role for SFRP1 as a suppressor in oral cancer (Figure 2b,c). Afterwards, immunohistochemistry (IHC) staining was analyzed for a cohort comprising 506 cases of head and neck squamous cell carcinoma (HNSCC) and noncancerous tissues. Among the entire cohort, 378/400 (75.8%) cases were negative, while 121/499 (24.2%) cases were positive (Figure 2d).

Regarding the clinicopathologic parameters associated with SFRP1 expression in OSCC, a univariate factor analysis indicated that high SFRP1 expression remained an independent predictor associated with early T-stage and clinical stage (Figure 3a,b). Among those with T_1–2_, the SFRP1 expression level was significantly higher than in the T_3–4_ samples (*p* = 0.004, 4.14 ± 2.08 versus 3.38 ± 2.08). Meanwhile, SFRP1 also had a higher expression in stage I-II tumors than in stage III-IV cases (*p* = 0.004, 4.23 ± 1.98 versus 3.45 ± 2.13). In contrast, there was no concordance between the SFRP1 expression level and N-stage, lymphovascular invasion, lymph node neck dissection and histologic grade from the TCGA data (*p* > 0.05) (Figure 3c–f). However, elevated SFRP1 expression was found among cN_+_ patients rather than in cN_0_ samples, which indicated that SFRP1 expression may be a potential sign of lymph node metastasis in OSCC patients.

### 3.2. Prognostic Signature of SFRP1 Expression

The correlation of SFRP1 expression with overall survival (OS) was assessed using a univariate survival analysis of 328 samples from TCGA (HR = 0.69, *p* = 0.024, Figure 4a). Additionally, similar results were obtained for disease-specific survival (DSS) and progress-free interval (PFI) after accounting for covariates (HR = 0.66, *p* = 0.047; HR = 0.66, *p* = 0.014, Figure 4b,c), which revealed that SFRP1 could have a substantial impact on the incidence of mortality and recurrence in OSCC patients. The findings also observed that SFRP1 expression was associated with clinical subtypes such as tongue squamous cell carcinoma (TSCC, HR = 0.47, *p* = 0.018, Figure 4e), while there was no obvious difference in SFRP1 expression for oral cavity cancer (*p* = 0.102, Figure 4d), implying that SFRP1 could be a viable prognostic indicator molecule in TSCC. Meanwhile, the area under the curve (AUC) and the value of the receiver operating characteristic (ROC) curve was 0.79, which emphasized the finding’s promising discriminative power in identifying tumors from normal tissues (Figure 4f).

The clinical characteristics and risk scores were then used to create a nomogram that integrated survival probability via multivariate analysis. We evaluated the 1-, 3- and 5-year survival probability regarding T-stage, N-stage, clinical stage, histological grade and SFRP1 expression level (Figure 4g). The higher the sum of the points calculated in a range from 0 to 100 was, the worse the prognosis was. It was found that cN_2_, histological G3 grade and low SFRP1 expression were risk factors for poor prognosis (*p* = 0.024, *p* = 0.048, *p* = 0.025). As expected, the calibration plot for the prediction of 3- and 5-year survival indicated that the observation and prediction were consistent (Figure 4f).

### 3.3. SFRP1 Is Associated with Tumor Immune Infiltration

Tumor immune infiltration was found to be associated with lymph node status, independently predicting the sentinel lymph node status and survival of OSCC [12]. This immune infiltration research suggested an active immune microenvironment (Figure 5a), and the expression of SFRP1 was positively correlated with the infiltration of mast cells, T-helper cells, eosinophils cells, T cells and Th1 cells (*p* < 0.001, Figure 5b–f).

### 3.4. Functional Analysis of SFRP1

Among the 668 significantly DEGs, 2 (0.30%) down-regulated genes and 666 (99.97%) upregulated genes were explored using GSEA (Figure 6a), with the adjusted *p*-value < 0.05 and the |log2(FC)| > 1.5 as the threshold. The 10 most significant expressed genes are shown in a heatmap (Figure 6b). SFRP1 and the correlated genes were subjected to GO and KEGG enrichment analyses. PPI network analysis was carried out to explore the relationships between SFRP1 and other genes in the DAVID database. The co-expressed genes included CTNNB1, FZD1, LRP5, LRP6 and WNT8B, most of which were associated with the Wnt signaling pathway (Figure 7a). Notably, BP-enriched terms were strongly associated with the regulation of membrane potential, postsynaptic membrane potential and synaptic transmission; the MF category included passive transmembrane transporter activity and ion-gated channel activity; the CC category was positively associated with the intrinsic and integral components of the synaptic membrane (Figure 7b–d).

Furthermore, GSEA was performed to compare high-SFRP1 and low-SFRP1 groups to identify related signaling pathways. The top eight signaling pathways with significant enrichment in the high-SFRP1 expression phenotype are shown in Figure 8 and include the calcium signaling pathway, neuroactive ligand–receptor interaction, the core matrisome, ECM glycoproteins and the anti-inflammatory response pathway, which may contribute to the interaction of pathophysiological mechanisms of SFRP1 in OSCC.

### 3.5. SFRP1 Represses Invasion and Migration of Cal27 Cells In Vitro

In view of the strong association of SFRP1 expression with OSCC, it is worth discussing the biological role of SFRP1 through in vitro experiments. Following the lentiviral transfection, knockdown and overexpression of SFRP1 in Cal27 cells that were prepared in vitro, we examined SFRP1 levels using both Western blotting and RT-PCR, suggesting a high efficiency of lentiviral transfection (Figure 9a,b).

As shown in Figure 9c,d, the transwell migration assay was utilized to detect if SFRP1 suppressed the invasion behaviors of Cal27 cells. Collectively, it was demonstrated that SFRP1 overexpression inhibited the invasion of Cal27 cells, whereas a great deal of Cal27 cells in the knockdown and control groups migrated to the lower chamber (*p* < 0.001). A wound-healing assay was performed to reveal the role of SFRP1 in OSCC migration. Similarly, SFRP1 overexpression resulted in a drastic decrease in migration ability when compared to the control group, but those with the knockdown of SFRP1 displayed an increase in migration behavior (*p* < 0.001, Figure 9e), indicating that the invasion and migration were significantly suppressed by the exogenous overexpression of SFRP1 in Cal27 cells.

### 3.6. SFRP1 Regulates Cell Cycle Progression and Suppresses Cell Proliferation In Vitro

Flow cytometry was performed to analyze the influence of SFRP1 on cell progression. In the SFRP1 knockdown group, the G0 + G1 phase percentage decreased from 31.18% to 18.90%; the proportions of cells in S-phase decreased from 52.02% to 36.48%. However, the overexpression of SFRP1 resulted in an accumulation of cells in the G0 + G1 phase, with the percentages of cells increasing from 20.16% to 41.95% (*p* < 0.001), while the number of S-phase cells did not vary significantly after the SFRP1 overexpression. Thus, the overexpression of SFRP1 probably induced G1 phase arrest and inhibited DNA replication (Figure 9f).

To further explore the instinct mechanism, the 3, 4 and 5 day proliferative rates of SFRP1 knockdown Cal27 cells were remarkedly higher than those in the control group according to the CCK8 assay (*p* < 0.001). Compared with scrambled control cells, the SFRP1 overexpression Cal27 cells showed a significantly low proliferative rate from the second day of the assay (*p* < 0.001), confirming the role of SFRP1 suppression in controlling proliferation (Figure 9g).

## 4. Discussion

Although traditional therapeutic strategies for OSCC such as surgery, chemotherapy and radiotherapy have significantly improved in recent years, the limitations of early diagnosis and locoregional relapse remain major challenges. Exploring molecular mechanisms is generally considered meaningful to guide clinical diagnosis and oncological outcomes. Subsequent integrative molecular analyses have revealed, for the first time, the important role of SFRP1, a well-characterized member of the Secreted Frizzled-Related Protein family, which has been verified to be a tumor suppressor in various neoplasms. The down-regulation of SFRPs has been observed in a series of solid tumors that contain colorectal cancer [13], gastric cancer [8], oropharyngeal cancer [14] and leukemia [15], which was also validated by our pan-cancer analysis. This research creatively integrated microarray data of SFRP1 with large OSCC samples from the TCGA database and in vitro experiments.

In this work, SFRP1 was found to be significantly down-regulated in clinical OSCC samples from public databases. Patients with low SFRP1 expression had a worse OS, DSS and PFI than patients with high expression based on the survival analysis. Moreover, low SFRP1 expression was not only positively related to advanced TNM stage, T stage and oral tongue subtypes, but also directly related to a worse survival rate, implying that SFPP1 could be a helpful biomarker for predicting diagnosis and prognosis in patients with OSCC. A nomogram multivariate analysis also indicated that down-expressed SFRP1 could be a potential predictor of 3- to 5- year survival as well as N-stage and high-histological grade. It also displayed a trend towards preferential expression in lymph node metastases. In line with the work of Chakraborty et al., the frequency of alterations for SFRP1 was significantly higher in lymph node positive cases (*p* = 0.025), which is in accordance with the reduced expression seen in the tumor group [16].

On the other hand, the epigenetic silencing of Wnt pathway antagonists has been well documented in the research of guiding tumorigenesis. Studies in this rapidly emerging field have revealed that the activation of the Wnt pathway may induce epithelial–mesenchymal transition (EMT) and mediate the tumor microenvironment by altering extracellular matrices, fibrotic processes and immune responses in HNSCC cells [17,18,19]. It is hypothesized that SFRPs contain a cysteine-rich homologous domain and may compete with Frizzled receptors for binding to Wnt ligands to produce antagonism. Thus, SFRP genes are called the “gatekeeper gene” of the Wnt pathway [20]. In HNSCC, it was revealed that SFRP1 and SFRP5 were methylated in OSCC patients who like chewing betel quid, while this was not observed in normal oral mucosa and precancerous lesions, suggesting that methylation changes were specific and sensitive to the carcinogenesis of OSCC [14]. HPV infection was strongly relevant with HNSCC, and it was reported to cause approximatively 5% of OSCC cases worldwide [21]. A previous work showed that the promoter hypermethylation SFRP1 was found in HPV-infected tissues or in cases with tobacco habits, with a downregulation of the Wnt pathway downstream target gene DKK1 and the up-regulation of LRP6 [16]. Certain limitations were noted in our study. Additional clinical factors such as HPV infection should be included to improve clinical application. Based on the above evidence and our microarray data, we propose that SFRP1 plays a key role in the occurrence and prognosis of OSCC through its molecular functions and epigenetic alterations.

The tumor microenvironment plays a vital role in both the positive and negative regulators of cancer hallmarks. An immunohistochemical study demonstrated that tumors infiltrated by tumor-infiltrating lymphocytes were relevant to producing better outcomes in HNSCC patients [22]. In this work, the positive correlation between SFRP1 and immune cells suggested that it could be one of the related genes that affected the tumor microenvironment of OSCC, especially in terms of the abundances of mast cells, T-helper cells, eosinophils cells, T cells and Th1 cells. Patients with relatively high SFRP1 expression levels could have a more effective response to immunotherapy, which may be one of the explanations for SFRP1 being shown to have value in predicting cancer progression and prognosis in our analyses. Previous studies have reported that low-levels of tumor infiltration in tumor tissues resulted in worse outcomes of cancer patients, including in CD4^+^ T cells and for macrophages in gastric cancer [23], as well as for B cells in breast cancer [24]. Interestingly, in the context of HNSCC, the expression signature of macrophages showed a worse prognosis for patients, while low levels of infiltrating cells showed a strong correlation with HPV infection in HNSCC [25]. This discrepancy could be attributed to autophagy-mediated immune escape from tumors in different tumor-immune cells [22]. Therefore, further research is required to elucidate the interaction network between SFRP1 and infiltrating immune cells.

In addition, co-expression genes and enrichment pathways were used to evaluate the comprehensive features and biological functions of SFRP1 in OSCC. With the high and low SFRP1 expressions as signatures of sample subgroups, the single gene enrichment analysis showed that SFRP1 participated in a wide range of metabolic cascades and transduction pathways, including the calcium signaling pathway, neuroactive ligand receptor interaction, the core matrisome, ECM glycoproteins, anti-inflammatory response pathways, etc., which suggested that SFRP1 expression was involved in the positive regulation of enrichment signaling cascades and SFRP1 might play a pathological role in biological oxidations, neuroactive ligand–receptor interaction and anti-inflammatory responses. However, each pathway does not exist in an isolated state; they form a complicated network with each other. Therefore, those related pathways do not deserve attention for their biological characteristics also. The cross-regulation of various cascades in which SFRP1 participates appear to offer opportunities for clinical treatments and subsequent research.

Since previous studies have confirmed SFRP1 to be an important regulator of cancer cell motility in skin tumor and colorectal cancer cell lines [26,27], few studies have reported the cell characteristics in OSCC. To achieve effective inhibition and overexpression in our study, we took advantage of adenovirus vectors to knock down and overexpress the target gene. Cancer is regulated by uncontrolled cell proliferation induced by the aberrant activity of cell-cycle-related genes [28]. In biological functional experiments, SFRP1 overexpression rendered the Cal27 cells more motile and invasive. Conversely, the knockdown of SFRP1 enhanced migration, proliferation and tumorigenesis. In regard to cell cycle, the depletion of SFRP1 expression resulted in G0+G1 phase arrest and the suppression of OSCC cell proliferation. Although the proliferation index (PI) of the SFRP1 knockdown group was relatively higher than that of the control group, we stumbled upon a high proportion in the G2+M phase, which caused an abnormity of Cal27 cell cycle transition. In summary, our results suggested that the overexpression of SFRP1 might affect the aggressiveness of Cal27 cells and might be a strategy for the targeted treatment of OSCC. One limitation should be noted in the lack of in vivo experiments, which are necessary for further verification to establish SFRP1 as a druggable target.

Indeed, a number of clinical trials of suppressors in malignancies have been initiated recently. Epigenetic modification is primarily mediated by DNA methyltransferases (DNMTs) and histone deacetylases (HDACs). The HDAC inhibitor, sodium butyrate, restored histone modifications in the promoter regions of SFRP1, two genes found in gastric cancer cells [29]. Sudha et al. found that the inhibition of SFRP4 could be restored by adding Wnt ligands (such as Wnt3a). SFRP4 could inhibit cancer stem-like cells (CSCs) from head and neck cancers through self-renewal, cloning and expression; moreover, CSCs with such biomarkers were more sensitive to the chemotherapeutic drug cisplatin [30]. Therefore, selecting an appropriate targeting gene will be key in tumor-targeted therapy and multidisciplinary team involvement.

## 5. Conclusions

At present, microarray data analyses and biological studies of suspicious genes are implemented for the adjuvant therapy of OSCC. In this study, we mined large databases to analyze the function, expression enrichment and immune infiltrating of SFRP1, with the aim of verifying the potential oncogenic and prognostic values of SFRP1 in OSCC. SFRP1 was overexpressed in normal tissues compared to in tumor tissues, especially in patients in the early clinical stage and early T stage. The function of SFRP1 was further verified in cytological experiments, where SFRP1 inhibited cell invasion, migration and proliferation and blocked cell cycle transition from G0 + G1 to S phase in Cal27 cells. While our study unraveled the inhibit function of SFRP1 in oral squamous cancer cells, it is necessary to expose a deeper mechanistic understanding of SFRP1-dependent signaling pathways, which could contribute to a cancer-specific single agent or combination therapy for OSCC.

## Figures and Tables

**Figure 1 biomolecules-12-01034-f001:**
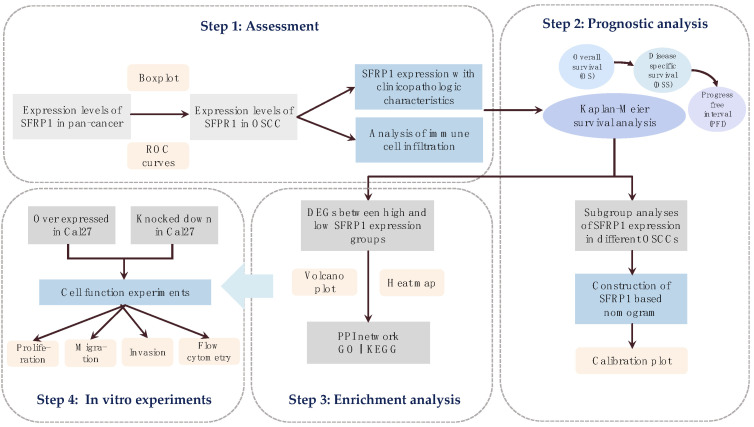
Flow chart of data analysis and experimental design.

**Figure 2 biomolecules-12-01034-f002:**
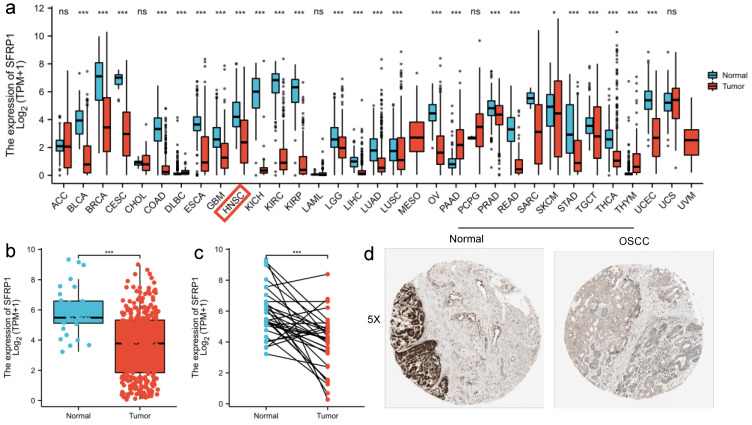
SFRP1 expression between cancer and normal tissues. (**a**) Expression levels of SFRP1 in solid tumors and adjacent noncancerous tissues containing 33 tissues from TCGA. The red box shows expression level of SFRP1 in head and neck cancers (* *p* < 0.05, *** *p* < 0.001). (**b**) Boxplot of SFRP1 expression levels in OSCC and matched normal tissues (*p* < 0.001). (**c**) Quantification of SFRP1 IHC staining in OSCC and matched normal tissues (*p* < 0.001). (**d**) Representative images of IHC in oral cancer tissue and control tissue.

**Figure 3 biomolecules-12-01034-f003:**
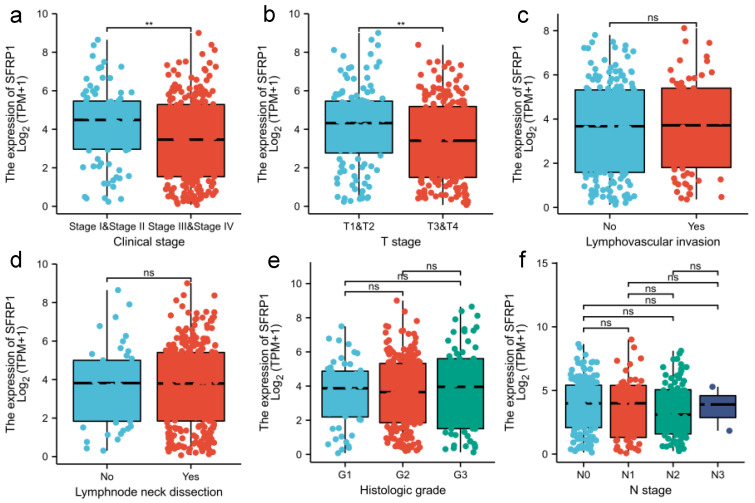
Association of SFRP1 expression with clinicopathologic characteristics. (**a**) Clinical stage; (**b**) T stage; (**c**) lymphovascular invasion; (**d**) lymph node neck dissection; (**e**) histologic grade; (**f**) N stage. All data are represented by mean ± SD and ns means non significance, ** *p* < 0.01.

**Figure 4 biomolecules-12-01034-f004:**
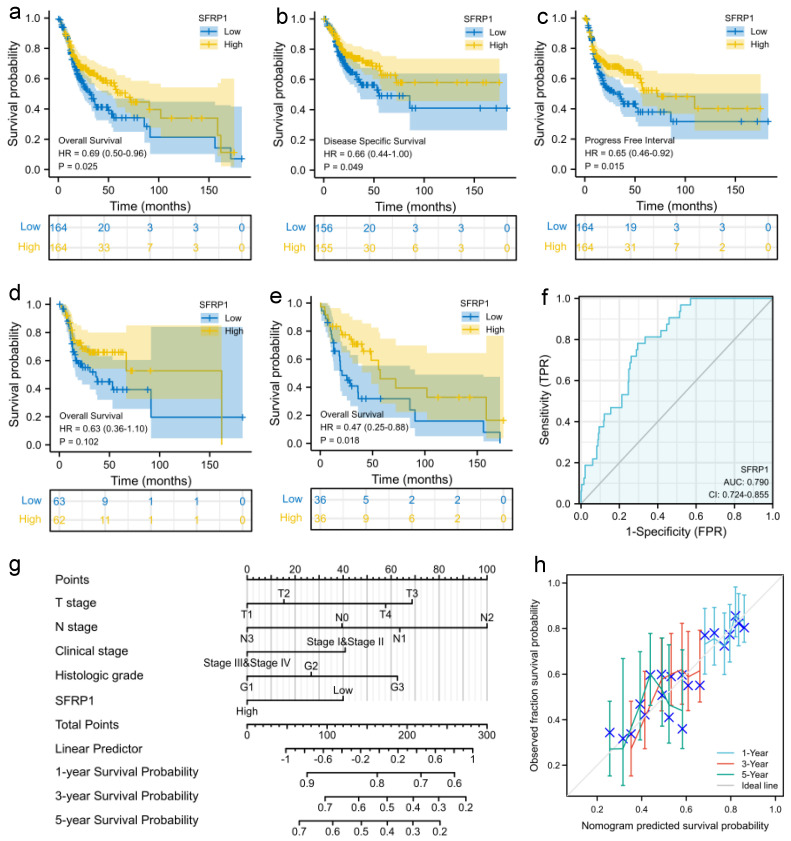
Prognostic analysis of various clinicopathological factors. (**a**) Kaplan–Meier survival curves in OS; (**b**) Kaplan–Meier survival curves in DSS; (**c**) Kaplan–Meier survival curves in PFI; (**d**) Kaplan–Meier survival curves in oral cavity cancer; (**e**) Kaplan–Meier survival curves in TSCC; (**f**) ROC curve showed that SFRP1 was a marker to predict the prognosis of OSCC; (**g**) nomogram used to predict the probability of 1-, 3- and 5-year OS for OSCC patients; (**h**) calibration plot of the nomogram.

**Figure 5 biomolecules-12-01034-f005:**
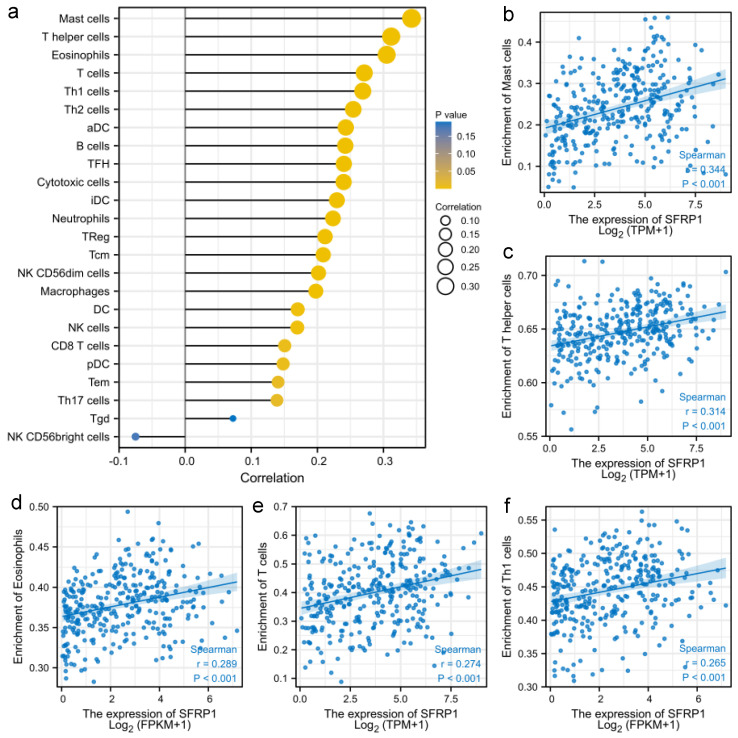
Relationship between SFRP1 and tumor immune infiltration in OSCC; (**a**) correlations between SFRP1 and relative abundance of 24 immune cells in OSCC; scatter diagram of the SFRP1 expression and enrichment of (**b**) mast cells, (**c**) T-helper cells, (**d**) eosinophil cells, (**e**) T cells, and (**f**) Th1 cells.

**Figure 6 biomolecules-12-01034-f006:**
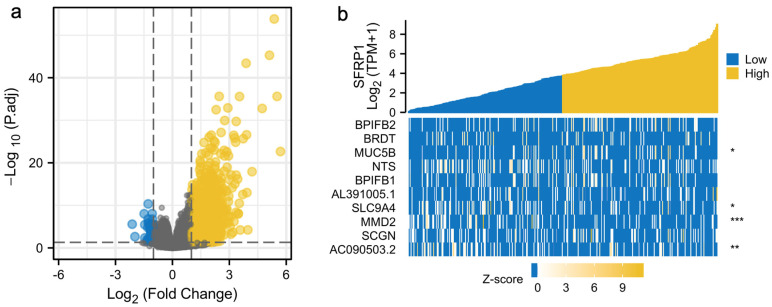
Differentially expressed genes (DEGs) between cases with high- and low-SFRP1-expression groups. (**a**) Volcano plot of differentially expressed genes; (**b**) heatmap of the top 10 significant differentially expressed genes. * *p* < 0.05, ** *p* < 0.01, and *** *p* < 0.001.

**Figure 7 biomolecules-12-01034-f007:**
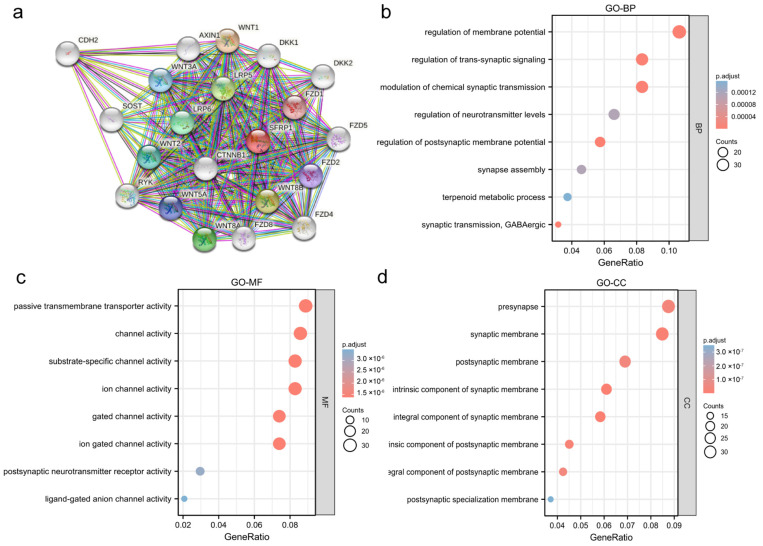
Protein–protein interaction (PPI) network and functional enrichment analysis. (**a**) PPI network of SFRP1 and co-expressed genes; (**b**) GO terms in the “biological process (BP)” category; (**c**) GO terms in the “molecular function (MF)” category; (**d**) GO terms in the “cellular component (CC)” category. Blue and red tones represent adjusted *p*-values and different circle sizes represent the number of DEGs.

**Figure 8 biomolecules-12-01034-f008:**
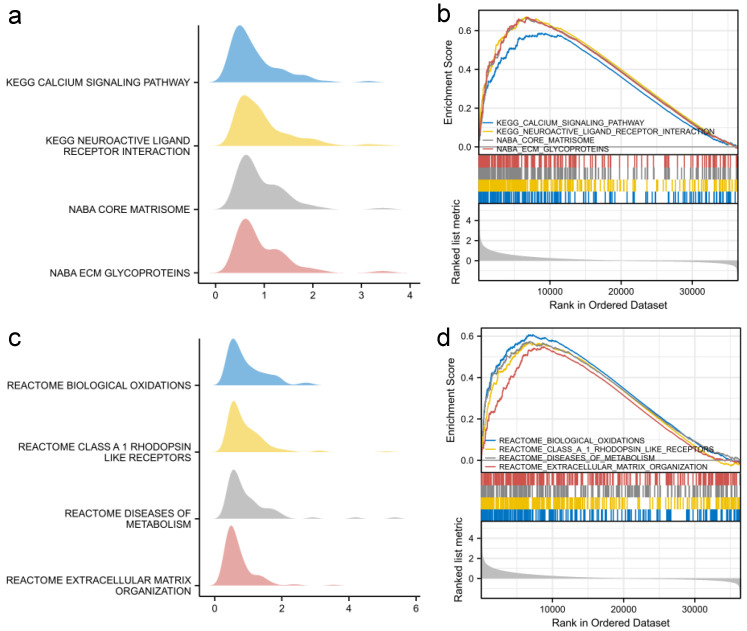
Enrichment plots of GSEA analysis. (**a**–**d**) Related eight main differentially enriched signaling pathways in SFRP1 high versus low samples.

**Figure 9 biomolecules-12-01034-f009:**
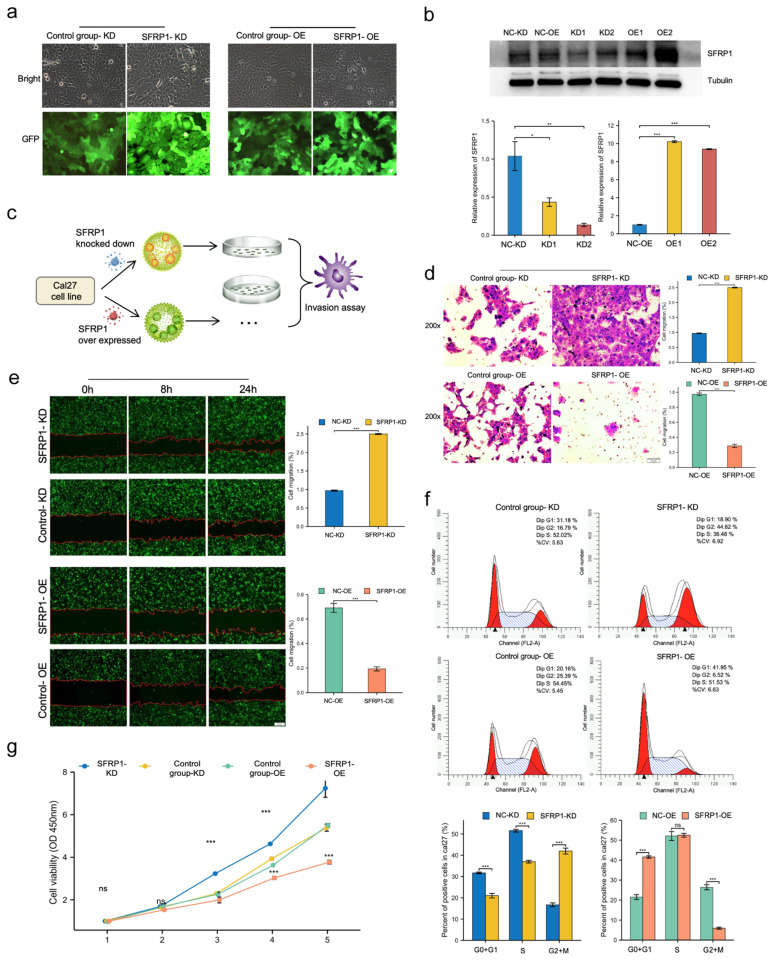
SFRP1 represses invasion, migration and proliferation of Cal27 cells in vitro. (**a**) SFRP1 was knocked down and overexpressed in Cal27 cells by lentivirus transfection. (**b**) Upper, Western blotting was used to verify SFRP1 protein level; lower, RT-PCR was utilized to detect SFRP1 expression in Cal27 cells. (**c**) Flow chart of transwell assay. (**d**) Overexpression of SFRP1 inhibited the invasion of Cal27 cells through the chamber. (**e**) Wound-healing assay demonstrated that SFRP1 overexpression inhibited cell migration. (**f**) Flow cytometry analyses of cell cycles of SFRP1 overexpressed and knocked down cells and their respective negative control cells. (**g**) CCK8 assays were performed to detect Cal27 proliferation based on the absorbance at various points of the day at 450 nm. * *p* < 0.05, ** *p* < 0.01, *** *p* < 0.001.

## Data Availability

The data presented in this study are available on request from the corresponding authors.

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
