# Peer review of "Expression of Tumor Suppressor SFRP1 Predicts Biological Behaviors and Prognosis: A Potential Target for Oral Squamous Cell Carcinoma"

_biomolecules, 2022, doi:10.3390/biom12081034_

Round 1
Reviewer 1 Report
I have read with great interest the paper and I congratulate Authors. The topic is original.
In introduction section, the standard of care for OSCC should be clearly described. It should be stressed the importance of multidisciplinary management of toxicities and advances in research. Therefore, introduction would be enhanced by addition of references, such as PMID: 27933385, PMID: 25479896 to better contextualize the issue at hand in oncologic scenario.
Reviewer 2 Report
Chen et al., have put together a nice manuscript describing the tumor suppressor potential of SFRP1 in OSCC. The authors have used TCGA OSCC and normal patient RNA seq data and performed bioinformatics analysis to see the differential expression of SFRP1 across different tumors, between normal and tumor etc. They have further shown correlation of this gene with survival and other clinicopath factors in OSCC. Overal,l the manuscript is interesting, the figures are appropriate, the topic is important and the analysis of public dataset is correctly done. However, it is lacking mechanistic insight and a causative experimental data to suggest that SFRP1 does function as a tumor suppressor. Following are my major concerns,
1. Though authors have shown enrichment of Calcium pathway there is no experimental evidence in Knockdown or overexpressing cells to complement the bio-informatics findings.
2. Western blot is required for knockdown and overexpressing cells for confirmation on the protein level.
3. An in-vivo tumor formation assay in nude mice with either overexpressing or knockdown cells is required to show a causal relationship between SFRP1 overexpression and tumor progression/development.
4. The English language correction is required for the manuscript.
Round 2
Reviewer 1 Report
ok
Reviewer 2 Report
The revision has addressed my concerns. The manuscript should be accepted.